# PERSPECTRA: A SCALABLE AND CONFIGURABLE PLURALIST BENCHMARK OF PERSPECTIVES FROM ARGUMENTS

**Shangrui Nie♠♡, Kian Omoomi♣, Lucie Flek♠♡, Zhixue Zhao◇, Charles Welch♣**

♠ Bonn-Aachen International Center for Information Technology, University of Bonn, Germany
♡ Lamarr Institute for Machine Learning and Artificial Intelligence, Germany
◇ Department of Computer Science, University of Sheffield, United Kingdom
♣ McMaster University, Canada

snie@uni-bonn.de, omoomik@mcmaster.ca

## ABSTRACT

Pluralism, the capacity to engage with diverse perspectives without collapsing them into a single viewpoint, is critical for developing large language models that faithfully reflect human heterogeneity. Yet this characteristic has not been carefully examined within the LLM research community and remains absent from most alignment studies. Debate-oriented sources provide a natural entry point for pluralism research. Previous work builds on online debate sources but remains constrained by costly human validation. Other debate-rich platforms such as Reddit and Kialo[1] also offer promising material: Reddit provides linguistic diversity and scale but lacks clear argumentative structure, while Kialo supplies explicit pro/con graphs but remains overly concise and detached from natural discourse. We introduce PERSPECTRA, a pluralist benchmark that integrates the structural clarity of Kialo debate graphs with the linguistic diversity of real Reddit discussions. Using a controlled retrieval-and-expansion pipeline, we construct 3,810 enriched arguments spanning 762 pro/con stances on 100 controversial topics. Each opinion is expanded into multiple naturalistic variants, enabling robust evaluation of pluralism. We initialise three tasks with PERSPECTRA: opinion counting (identifying distinct viewpoints), opinion matching (aligning supporting stances and discourse to source opinions), and polarity check (inferring aggregate stance in mixed discourse). Experiments with state-of-the-art open-source and proprietary LLMs, highlight systematic failures, such as overestimating the number of viewpoints and misclassifying concessive structures, underscoring the difficulty of pluralism-aware understanding and reasoning. By combining diversity with structure, PERSPECTRA establishes the first scalable, configurable benchmark for evaluating how well models represent, distinguish, and reason over multiple perspectives. We release PERSPECTRA as a resource with flexible configurations, enabling the creation of tasks beyond the demo tasks presented in this paper, and fostering progress toward pluralism-sensitive systems that more faithfully capture human heterogeneity. The dataset is available on github page[2].

## 1 INTRODUCTION

Large language models (LLMs) are increasingly deployed in settings that serve heterogeneous user communities, from education and healthcare to content moderation and public deliberation. Because these users hold divergent values and perspectives, it is essential that LLMs outputs are not homogenized into a single "average" answer but instead reflect the plurality of legitimate viewpoints. Yet

---

[1] https://www.kialo.com/
[2] https://github.com/caisa-lab/ICLR-2026-Pespectra

current models often converge on narrow, homogenized answers, which risks obscuring the diversity of perspectives present in real-world discourse. Ensuring inclusive and trustworthy deployment of LLMs therefore requires benchmarks that explicitly evaluate models' ability to represent and distinguish multiple perspectives.

Progress toward this goal has been supported by emerging pluralism-oriented datasets. For example, OpinionQA Santurkar et al. (2023) and GlobalOpinionQA Durmus et al. (2024) align model predictions with survey responses to test whether models capture population-level distributions of opinion. The DICES dataset Aroyo et al. (2023) collects culturally diverse judgments of conversational safety. While such resources highlight the feasibility of pluralism-aware evaluation, they depend heavily on human annotation or curated surveys. As a result, they are costly to expand, narrow in topical scope, and difficult to adapt to new pluralistic tasks.

An alternative line of research looks to naturally occurring debates. The PERSPECTRUM dataset (Chen et al., 2019) demonstrates the value of online debate sources for capturing diverse perspectives, but its scalability is constrained by intensive human validation. Other debate-rich platforms offer complementary benefits: Reddit provides scale and linguistic variety, reflecting the informality of real discourse (Pougué-Biyong et al., 2021), while Kialo supplies explicit pro/con graphs that enable resources such as KialoPrime (Sahitaj et al., 2024). Yet Reddit lacks argumentative structure and is costly to annotate, and Kialo's concise, formalized claims fail to mirror the stylistic richness of authentic discussions.

We address this gap by introducing the PERSPECTRA, a dataset that integrates the clarity of Kialo's debate structure with the diversity of Reddit discourse. Our construction pipeline retrieves semantically related Reddit comments for each Kialo opinion and uses controlled prompting with ChatGPT-4o to generate expanded arguments that remain faithful to the original stance while adopting the linguistic richness of authentic online discussions. This process yields a scalable resource of 3,810 expanded arguments across 762 pro/con opinions on 100 controversial topics.

The dataset naturally supports multiple pluralism-relevant evaluation tasks. We formalize three benchmarks: opinion counting (estimating the number of distinct opinions in a paragraph), opinion matching (aligning expanded arguments to their original claims), and polarity check (inferring aggregate stance from mixed arguments).

---

Games have always been more about playability and commercial success than being classified as art. It's crucial to acknowledge both the artistic merit and the potential health hazards these games can present. Whether it's the sweeping landscapes of an open-world RPG or the detailed character designs in a narrative-driven game, these elements come together to form a cohesive piece of art that deserves recognition. Just because you love a game doesn't magically turn it into a masterpiece of art; it's still a commercial product at its core. That's what art is all about—bringing ideas to life in a way that moves and inspires people. Maybe it's just a matter of time before a game comes along that changes everything, but for now, their impact remains largely within the gaming community.

---

Table 1: Example input to the opinion counting task. Sentences are color-coded by opinion cluster; expressions of the same opinion may appear in non-adjacent positions. The example illustrates the challenge posed to models: they must navigate interleaved opinions and group sentences that share the same underlying viewpoint despite lexical and stylistic variation.

## 2  RELATED WORK

### 2.1  PLURALISM AND PERSPECTIVISM IN CURRENT LLMS

LLMs often fail to reflect the diversity of viewpoints present in human discourse, instead converging on a normative "average." Alignment methods such as reinforcement learning from human feedback (RLHF) optimize for a single response style but in doing so suppress heterogeneity of opinions (Sorensen et al., 2024; Kirk et al., 2023). Empirical studies confirm this: RLHF narrows distributional pluralism by concentrating probability mass on a few answers in survey-style benchmarks (Santurkar et al., 2023; Durmus et al., 2024), reduces diversity across tasks (Kirk et al., 2023),

and leads to more homogeneous outputs in co-writing settings (Padmakumar & He, 2024; Slocum et al., 2025). Further analyses show that alignment can bias models toward majority or culturally dominant perspectives, effectively silencing minority or contrarian views (Chakraborty et al., 2024; Shahid et al., 2025; Sourati et al., 2025). These limitations have motivated interest in pluralistic alignment: Sorensen et al. (2024) propose frameworks such as Overton, steerable, and distributional pluralism to preserve viewpoint diversity. In summary, alignment improves safety and coherence but risks undermining the pluralism and perspectivism essential for inclusive AI. This gap underscores the need for benchmarks and datasets that test whether models can represent multiple perspectives.

## 2.2 DATASETS AND BENCHMARKS FOR PLURALISTIC MODELING

Pluralism-aware NLP has been advanced through new datasets and evaluation frameworks that capture diverse perspectives and value judgments. Building on Sorensen et al.'s taxonomy of pluralism (Sorensen et al., 2024), subsequent resources demonstrate both the promise and the limitations of current approaches. OpinionQA (Santurkar et al., 2023) aligns model outputs with U.S. opinion surveys, while GlobalOpinionQA (Durmus et al., 2024) extends this to cross-cultural settings. To capture divergent safety norms, Aroyo et al. (2023) introduce DICES, and CoSApien (Zhang et al.) focuses on safety pluralism through expert-defined configurations. Beyond human annotation, synthetic methods leverage LLMs to generate diverse perspectives. Hayati et al. (2024) show that prompting can elicit a wide range of moral and social viewpoints, offering scalable alternatives to multi-annotator pipelines (Röttger et al., 2022). Other frameworks include Modular Pluralism (Feng et al., 2024), which composes community-specific LLMs with a base model to represent perspectives without retraining. Relatedly, Community Alignment (Zhang et al., 2025) and MoralBench (Chiu et al., 2025) target end-state and process-level preference pluralism but are primarily used as training corpora rather than structured multi-viewpoint evaluation benchmarks. Recent surveys consolidate these directions (Xie et al., 2025), but emphasize open challenges: pluralistic datasets remain narrow in topical scope, curating "valid" perspectives is difficult, and evaluation must balance viewpoint coverage with coherence.

## 2.3 ARGUMENT MINING AND DEBATE CORPORA

Debate platforms provide natural ground for studying pluralism, but existing corpora present trade-offs. Kialo's structured pro/con trees support resources such as KialoPrime (Sahitaj et al., 2024) and BERDS (Chen & Choi, 2025), which enable fine-grained relation classification and perspective diversity evaluation, while PERSPECTRUM (Chen et al., 2019) demonstrates the value of curated perspectives for claims. Beyond these, DebateSum (Roush & Balaji, 2020) and OpenDebateEvidence (Roush et al., 2024) offer large-scale corpora derived from competitive formal debates. Their hierarchical claim–evidence structure and stance alignment support summarization and retrieval tasks at scale, yet remain focused on binary pro/con framing and formal styles. Similarly, IBM's Project Debater (Slonim et al., 2021) introduced tools for claim detection, evidence-based summarization, and automated speech generation, but aims to synthesize persuasive monologues rather than preserve multiple viewpoints. Yet these datasets remain limited in topical coverage and often collapse debates into binary pro/con stances; moreover, resources like PERSPECTRUM (Chen et al., 2019) rely on costly human annotation, which hinders scalability. Reddit, by contrast, offers large-scale, naturalistic discourse as in ChangeMyView (CMV) corpus (Gurjar et al., 2025) and DEBAGREEMENT (Pougué-Biyong et al., 2021), but annotation is also costly and labels often noisy due to ambiguity and sarcasm. Other resources such as the Internet Argument Corpus 2.0 (Abbott et al., 2016) similarly compile large-scale online debates from political forums with annotations but exhibit considerable lexical noise and lack explicit argument scaffolding. In summary, existing debate-derived datasets either provide clean structure at small scale or noisy diversity at large scale, highlighting the need for more scalable resources that combine both. Our work addresses this gap by integrating the structural clarity of Kialo with the linguistic diversity of Reddit into a scalable benchmark.

## 3 DATASET CONSTRUCTION

We construct a dataset that integrates the structural clarity of Kialo debates with the linguistic diversity of Reddit discussions. As illustrated in Figure 1, the pipeline begins with topic–opinion pairs from Kialo, retrieves semantically related Reddit comments, and then expands each (topic, opinion,

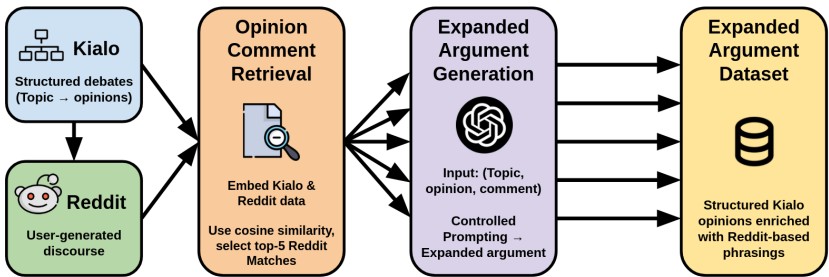

Figure 1: Overview of the dataset construction pipeline. Kialo debates (topics and opinions) are paired with Reddit comments via retrieval, then expanded through controlled prompting to produce naturalistic argument variants. The resulting dataset contains structured opinions enriched with Reddit-based phrasings.

comment) triple into a naturalistic argumentative statement via controlled prompting. The resulting collection consists of structured Kialo opinions enriched with Reddit-based phrasings. In the following subsections, we describe the data sources, retrieval procedure, generation method, and dataset statistics.

## 3.1 DATA SOURCES

Our dataset construction relies on two complementary sources. `Kialo` provides structured debate content, organized into topics and their associated pro/con opinions. Each topic corresponds to a central question or claim, while opinions represent user-submitted arguments supporting or opposing that topic. This hierarchical structure offers a clean and well-defined set of argumentative units that can serve as anchors for data generation. An example from Kialo is as follows:

> **Topic:** All people in the US should have the right to basic healthcare
> **Pros:** Healthcare for all people would save many lives in the US.
> **Cons:** A doctor shortage could occur if everyone has healthcare.

In contrast, Reddit supplies large-scale user-generated comments from diverse online discussions. Reddit threads are not organized into explicit debate trees but contain rich, informal argumentative expressions. These comments are used to provide contextual variation and linguistic diversity to the structured Kialo opinions.

By combining these two sources, we obtain both a reliable backbone of clearly formulated opinions and a wide pool of naturally occurring discourse from which additional context can be drawn.

## 3.2 OPINION–COMMENT RETRIEVAL

To ground structured Kialo opinions in naturally occurring discourse, we constructed for each topic a pool of candidate Reddit comments. Relevant threads were retrieved using search-based ranking to prioritize semantically related discussions. Importantly, there was no fixed set of subreddits, and threads could originate from any subreddit. To maintain topical diversity, we enforced a fixed retrieval budget per thread rather than relying heavily on a single source. For each topic, we retrieved the 20 most relevant threads. From each thread, we collected at most 100 comments, ranked by score (number of upvotes minus downvotes). If a thread contained fewer than 100 comments, all available comments were included. This process ensured that, in the maximum case, each topic contributed up to 2,000 comments (20 threads × 100 comments). This processed yielded an initial pool of raw candidates.

A multi-stage filtering pipeline was then applied. Comments containing fewer than five words were discarded, as they typically lacked argumentative substance. Additional filters removed low-quality or non-user content, including comments without an identifiable author, comments formally distin-

guished as moderator posts, comments authored by "automoderator," and comments with usernames containing "bot". Through this process, each topic yielded a candidate pool of roughly 1,500–1,800 comments.

For opinion–comment matching, both the original Kialo opinion and all candidate comments were embedded using the Qwen3-Embedding-8B model. We selected this model due to its strong clustering performance on the Huggingface MTEB leaderboard,[3] which is particularly relevant given that our task is semantic clustering between concise structured opinions and linguistically diverse Reddit discourse. Relevance was computed via cosine similarity, and for each opinion we retained the top five comments as semantically distinct matches. The highest-scoring candidate was designated as the primary "best match," while the remaining top-$k$ served as additional material for generating stylistically and contextually varied expansions.

### 3.3 EXPANDED ARGUMENT GENERATION

Each selected opinion–comment pair then serves as input to the expansion stage. For every triple of (topic, opinion, comment), we prompt GPT-4o with a fixed template designed to preserve the core stance of the original Kialo opinion while adapting its style and elaboration using cues from the paired Reddit comment. The prompt explicitly instructs the model to (i) maintain argumentative fidelity, (ii) ignore irrelevant or off-topic content, and (iii) mimic the informal and contextually grounded style of Reddit discussions. After piloting several prompt variants, we adopted the version that most reliably balanced fidelity and naturalness (see Appendix D).

This process results in five expanded variants for every opinion, producing a total of 3,810 expansions across 762 opinions. On average, each generated argument is about 100 words long, yielding a corpus of several hundred thousand words of enriched discourse. These multiple expansions capture alternative phrasings and contextual enrichments of the same underlying claim, substantially increasing the linguistic diversity of the dataset.

### 3.4 HUMAN ANNOTATION

The manual check criteria were designed to cover the major aspects of data quality that are particularly relevant to our construction pipeline. In practice, we found it useful to think of the evaluation in terms of three complementary layers.

**Content fidelity and coherence** The most fundamental requirement is that the expanded argument should remain faithful to the original opinion. Fidelity serves as the primary indicator of dataset quality: if the expansion drifts away from or even contradicts the source, the data loses its intended purpose. We therefore focus on whether the generation preserves the core stance and reasoning of the original opinion, and whether any added material stays on-topic rather than introducing unrelated or exaggerated claims.

**Mechanism-specific reliability** Our dataset construction relies on retrieving a best match Reddit post based on embedding similarity. However, semantic similarity alone does not guarantee that the retrieved content is truly relevant to the opinion or the topic. In our inspection, we indeed encountered cases where the best match was only loosely related, or even off-topic. To account for this, we distinguish between two aspects: the intrinsic relevance of the retrieved post itself, and the model's handling of that retrieval. A high-quality generation should selectively integrate arguments from a relevant match, or, if the match is irrelevant, demonstrate the ability to largely ignore it.

**Naturalness of Writing** Beyond correctness and retrieval, we require the data to resemble authentic discourse. The naturalness of writing criterion captures this dimension. Our aim is that the expansions should plausibly read as human-authored posts, consistent with the style of the original Reddit discussions. During inspection, we noted that certain outputs, although content-wise accurate, exhibited "machine-like" repetition or phrasing patterns. Such artifacts could undermine downstream realism and weaken the utility of the dataset.

---

[3] https://huggingface.co/spaces/mteb/leaderboard

Taken together, these three layers motivate five concrete evaluation criteria: fidelity to the original opinion, relevance of added content, relevance of the best match post to the topic, use of the best match post, and naturalness of the writing. Detailed definitions and the full 0–5 scoring rubrics are provided in Appendix C.

---

**Topic:** Marvel universe is better than DC universe.
**Opinion:** DC has better characters.
**Best Match:** DC's characters stand alone better, whereas Marvel does better team books
**Expanded:** While Marvel might excel at crafting epic team dynamics and ensemble stories, DC's strength lies in its iconic, standalone characters. Heroes like Batman, Superman, and Wonder Woman have rich, layered narratives that allow them to shine individually, creating a deep connection with fans. These characters have been developed with such depth and complexity that they can carry entire stories on their own, without needing a team to back them up. This ability to stand alone and still captivate audiences is what sets DC's characters apart and makes them truly legendary in the comic book world.
**Fidelity to Original Opinion**: 5
**Relevance of Added Content**: 5
**Use of Best Match Post**: 5
**Naturalness of Writing**: 4
**Relevance of Best Match to Topic**: 5

---

To make the evaluation procedure concrete, the above example shows one annotated case. The retrieved best match comment is highly consistent with the original opinion, and the expanded argument successfully integrates the cues from the comment (e.g., "standalone characters") while elaborating the stance with richer details. As a result, the expansion received perfect scores on fidelity, relevance, and match usage. However, annotators noted that the writing still retained minor traces of model-like phrasing, leading to a slightly lower score (4/5) on naturalness of writing.

## 3.5 STATISTICS

**Manual Evaluation of Data Quality**   Building on the manual check criteria defined above, we conducted a human evaluation to obtain quantitative scores for data quality. A random sample of 100 opinion–expanded pairs was annotated on the five dimensions using a 0–5 scale. The aggregated results are shown in Table 2. Overall, the expansions score highly on fidelity, added relevance, and naturalness, while the main source of variability lies in the topic relevance of the retrieved best match posts.

Table 2: Human evaluation results on 100 randomly sampled opinion–expanded pairs. Scores are on a 0–5 scale.

| Criterion | Mean | Std | Min | Max | Median | IQR (p25–p75) |
|---|---|---|---|---|---|---|
| Fidelity to Original Opinion | 4.92 | 0.40 | 2.0 | 5.0 | 5.0 | 5.0 – 5.0 |
| Relevance of Added Content | 4.87 | 0.46 | 2.0 | 5.0 | 5.0 | 5.0 – 5.0 |
| Use of Best Match Post | 4.72 | 0.65 | 2.0 | 5.0 | 5.0 | 5.0 – 5.0 |
| Naturalness of Writing | 4.42 | 0.48 | 4.0 | 5.0 | 4.0 | 4.0 – 5.0 |
| Relevance of Best Match to Topic | 3.31 | 1.37 | 0.0 | 5.0 | 3.0 | 3.0 – 5.0 |

**Corpus Summary**   In addition to these human evaluation results, we also report corpus-level statistics that capture the structural properties of the dataset as a whole. The dataset comprises 100 topics and 762 opinions (373 pros, 389 cons), yielding a near stance balance of 49.0% vs. 51.0% (Table 3a). Each opinion is expanded into five variants, totaling 3810 expansions. As shown in Table 3b, per-topic candidate sets are compact on average (mean 7.62 opinions; std 3.13) but display moderate heterogeneity (range 4–18), with pros and cons similarly distributed per topic (means 3.73 and 3.89; std 1.69). Text lengths align with the dataset's design: original opinions are short (mean 14.09 words; range 3–90), while expansions are length-stable around 100 words (mean 97.82; std

Table 3: Corpus-level and per-topic statistical summaries.

(a) Corpus-level summary of PERSPECTRA

| Metric | Value |
|---|---|
| # Topics | 100 |
| # Opinions (total) | 762 |
| # Pros / # Cons | 373 / 389 |
| Pro : Con (%) | 49 : 51 |
| # Expanded (total) | 3810 |
| Expanded per opinion | 5 (fixed) |

(b) Per-topic opinion counts and text length statistics

| Category | mean | std | min | max |
|---|---|---|---|---|
| **Per-topic opinion counts** | | | | |
| Pros per topic | 3.73 | 1.69 | 2 | 11 |
| Cons per topic | 3.89 | 1.69 | 2 | 8 |
| Total per topic | 7.62 | 3.13 | 4 | 18 |
| **Text length statistics (in words)** | | | | |
| Opinion | 14.09 | 9.02 | 3 | 90 |
| Expanded | 97.82 | 17.51 | 49 | 311 |

17.51; range 49–311), enabling predictable input budgeting for downstream tasks while preserving natural variation across topics.

## 4 Downstream Tasks Application

While PERSPECTRA itself consists only of expanded arguments and their original opinion, it naturally supports a range of evaluation tasks. These tasks are derived from the structural properties of the data, such as the mapping between opinions and their multiple expanded arguments, and the original stance of the original opinion. In this section, we formalize the primary downstream tasks that can be directly constructed from the PERSPECTRA.

**Opinion Counting**  We first apply PERSPECTRA to formulate a task of estimating how many distinct opinions are present within a paragraph. To construct inputs, we concatenate multiple expanded versions, some of which may originate from the same underlying Kialo opinion. While the expansions differ in surface realization, they should be treated as instances of the same opinion. As illustrated in Table 1, semantically equivalent arguments may appear in varied linguistic forms and are not necessarily adjacent within the paragraph. The task thus evaluates whether a model can abstract away from lexical variation and identify semantically consistent arguments that share a common stance.

Formally, given a paragraph $x$ consisting of $m$ expanded arguments, the model predicts an integer $\hat{y}$ corresponding to the number of unique underlying opinions. The ground truth $y$ is determined by the distinct Kialo opinion identifiers associated with the expansions. We report three metrics: (i) *Accuracy*, requiring exact match $\hat{y} = y$; (ii) *Mean Absolute Error (MAE)*, measuring average deviation $|\hat{y} - y|$; and (iii) *Normalized Inverse Error (NIE)*, a smoother score defined as NIE $= \frac{1}{N} \sum_{i=1}^{N} (1 + \frac{|\hat{k}_i - k_i|}{\max(1, k_i)})^{-1}$, where $N$ is the number of evaluation examples, $k_i$ is the ground-truth count of unique opinions in the $i$-th example, and $\hat{k}_i$ is the model prediction. The denominator uses $\max(1, k_i)$ to normalize the error relative to the scale of the true count.

**Opinion Matching**  The second task tests whether a model can correctly associate an expanded argument with its original source opinion. For each topic, we randomly select one expanded version and present it together with the full set of candidate opinions (both pro and con) defined under that topic. The model must identify which original opinion the expansion derives from.

Performance is measured with two complementary metrics. *Accuracy* requires exact identification of the source opinion, while *Stance Accuracy* considers only whether the chosen candidate belongs to the correct debate side (pro vs. con).

**Polarity Check**  The third task evaluates whether a model can infer the aggregate stance of a paragraph containing a mixture of expansions. For each topic, we randomly sample $n$ expanded versions and concatenate them into a single paragraph. The model must assign a binary label (pro or con) indicating the dominant orientation of the paragraph. The gold label is determined by majority

Table 4: Results of all three sub-tasks (T1–T3): Opinion Counting (Accuracy, MAE, NIE), Opinion Matching (Accuracy, Stance Accuracy), and Polarity Check (Accuracy) on models and human.

| Models | T1: Opinion Counting | | | T2: Opinion Matching | | T3: Polarity Check |
|---|---|---|---|---|---|---|
| | Acc. ↑ | MAE ↓ | NIE ↑ | Acc. ↑ | Stance Acc. ↑ | Acc. ↑ |
| LLaMA-3.1-8B-Instruct | 6.4% | 3.17 | 0.57 | 6.0% | 10.4% | 29.6% |
| Falcon3-7B-Instruct | 15.8% | 2.28 | 0.67 | 66.2% | 77.4% | 51.0% |
| Qwen3-8B | 33.6% | 1.26 | 0.79 | 83.6% | 92.4% | 64.0% |
| Qwen3-32B | 28.6% | 1.62 | 0.74 | 71.6% | 86.0% | 67.6% |
| Qwen2.5-7B-Instruct | 30.8% | 1.19 | 0.78 | 78.8% | 91.2% | 58.6% |
| Qwen2.5-32B | 35.2% | 1.25 | 0.79 | 72.2% | 80.4% | 55.2% |
| QwQ-32B | 36.2% | 0.97 | 0.82 | 72.0% | 80.2% | 66.4% |
| DS-R1-Distill-Llama-8B | 23.6% | 2.24 | 0.69 | 66.6% | 84.8% | 54.8% |
| DS-R1-Distill-Qwen-7B | 32.6% | 1.21 | 0.79 | 66.0% | 89.4% | 54.8% |
| DS-R1-Distill-Qwen-32B | 31.2% | 1.48 | 0.77 | 85.4% | 95.0% | 67.2% |
| GPT-4o | 29.2% | 1.38 | 0.76 | 74.8% | 81.2% | 76.4% |
| GPT-4o-mini | 34.0% | 0.94 | 0.81 | 71.6% | 81.6% | 72.8% |
| Human | 44.0% | 1.01 | 0.81 | 90.6% | 94.3% | 85.6% |

vote over the stance labels of the included expansions. Performance is assessed using *Accuracy*, computed as the proportion of paragraphs for which the predicted polarity matches the ground truth.

Although pluralism-related tasks inherently involve subjective perspectives, PERSPECTRA is designed to minimize such subjectivity in task formulation. All opinion units are inherited directly from Kialo's moderated debate graphs, where topic–opinion boundaries are externally curated rather than specified by us. To accommodate any remaining ambiguity, our evaluation relies on graded, distance-sensitive metrics (e.g., MAE and NIE) instead of brittle exact matches, ensuring proportional rather than all-or-nothing penalties.

## 5 RESULTS AND FAILURE ANALYSIS

### 5.1 OVERALL BENCHMARK RESULTS

For each of the three sub-tasks, we instantiated an evaluation set of 500 examples by sampling from the PERSPECTRA corpus. Since task-specific ground truths can be derived from the opinions and stance labels, this procedure requires no additional human annotation. Moreover, the construction process is fully programmatic and therefore easily scalable.

Table 4 summarizes results across the three evaluation tasks. Performance varies substantially across model families, with no single system achieving consistently high scores across all sub-tasks. For opinion counting (T1), the best results come from open-source models such as QwQ-32B, which slightly outperforms GPT-4o on both accuracy and NIE. For opinion matching (T2), distillation-based Qwen models (e.g., DS-R1-Distill-Qwen-32B) achieve the strongest accuracies, exceeding even GPT-4 family. By contrast, polarity check (T3) remains most challenging for open-source models, where GPT-4o leads with the highest accuracy. Overall, these findings confirm that pluralism-oriented tasks remain difficult: while some specialized models excel at particular sub-tasks, substantial headroom remains for models to achieve robust pluralistic reasoning across all tasks simultaneously. We additionally report human performance for all sub-tasks, which exceeds every evaluated model by a clear margin.

### 5.2 CHALLENGE 1: OPINION OVERESTIMATION

In the opinion counting task, we observe that models tend to overestimate the number of distinct opinions. This pattern suggests that **the primary challenge does not lie in detecting stance, but rather in correctly aggregating semantically similar expansions under the same original opinion**. In other words, the task is less about classification and more about avoiding over-splitting. A representative failure case is shown in Table 5 in the Appendix F, where a concatenated paragraph contains eight expanded versions. Despite the apparent diversity of expression, these expansions

actually derive from only four distinct original opinions. For example, four of the segments all rephrase the idea that schools enforce conformity in ways that harm students' mental health, while two segments both emphasize the safeguarding role of schools in identifying early signs of distress. The remaining two expansions represent distinct yet separate perspectives.

When confronted with such input, models typically predict a much higher count than the ground truth, possibly because they mistake surface-level lexical or stylistic differences for distinct opinions. To quantify these patterns, we further analyzed the subset of 47 inputs on which all 12 evaluated models failed, yielding a total of 564 incorrect predictions. Among these, 71.5% oversplit errors (predicting more opinions than the ground truth) and 24.6% undercount errors (predicting fewer). Only 3.9% consisted of invalid outputs. This distribution confirms that oversplitting dominates as the primary failure mode, consistent with the qualitative analysis above. In practice, models often mistake surface-level lexical or stylistic differences for distinct opinions, underscoring that the real difficulty of the task lies in semantic normalization: different expanded versions must be clustered into their underlying original opinion, rather than treated as independent contributions.

### 5.3 CHALLENGE 2: SEMANTIC OVERLAP IN MATCHING

In the opinion matching task, errors rarely come from confusing opposite stances. As shown in Table 4, stance accuracies are consistently 8–24 points higher than exact-match accuracies across models (e.g., Qwen3-8B reaches 83.6% accuracy but 92.4% stance accuracy; DS-R1-Distill-Qwen-7B achieves 66.0% vs. 89.4%). This gap indicates that **models generally succeed in identifying the correct side of a debate, but struggle to capture fine-grained distinctions between semantically related opinions on the same side**. Expanded versions often introduce additional details (e.g., specific cases, examples, or contextual nuances) that are not explicitly present in the concise original opinions, creating ambiguity between multiple candidate matches. For instance, in the topic "Online video games are currently more enjoyable than board games," one expanded opinion mentioned the immersive experience of online games, including global connectivity, rich storylines, and convenience. While the ground truth was "Online video games contain additional features that make them a more enjoyable experience than board games," the expansion also overlapped with other pro-side options such as "The story of a video game is more involved" and "You don't need to leave your room," leading to frequent confusion. Similarly, in the topic "Trophy hunting should be illegal," an expanded opinion emphasized that trophy hunting generates funds for conservation and reinvests in local communities. The correct match was "Trophy hunting helps conservation efforts," but the mention of local benefits made it easily confused with "Trophy hunting benefits local communities." These cases highlight that exact-match errors are primarily due to over-specificity and semantic overlap, while stance-level matching remains relatively robust.

### 5.4 CHALLENGE 3: CONCESSION TRAP IN POLARITY

A major source of **polarity errors is the concession–rebuttal structure**: texts that begin by acknowledging one side and then pivot to argue the opposite. Local polarity cues in the concessive lead ("while/although...") often mislead models toward the conceded side instead of the concluding stance. In the federal cannabis legalization samples, several expansions noted potential downsides such as health risks or high taxation, but then concluded with strong pro-legalization arguments (e.g., dismantling the black market, reducing incarceration, generating tax revenue). While the overall polarity was pro, models often misclassified such inputs as con because the initial concessions blurred the aggregate direction. To quantify this phenomenon, we examined the 34 inputs on which all models failed. Among them, 20 cases (59%) displayed a clear concession–rebuttal structure, confirming that polarity errors are predominantly systematic, arising from models being misled by concessive openings rather than from random misclassification.

## 6 CONCLUSIONS

We introduced PERSPECTRA, a pluralist benchmark that integrates the structural clarity of Kialo debates with the linguistic richness of Reddit discourse. Through a retrieval-and-expansion pipeline, the dataset provides 3,810 enriched arguments across 100 topics, each opinion expressed in multiple naturalistic variants, making the resource both scalable and easily configurable for diverse evaluation

settings. Using the dataset, we derived three sub-tasks: opinion counting, opinion matching, and polarity check. Across these tasks, we uncovered recurring failure modes. In particular, models exhibit systematic opinion overestimation driven by oversplitting in counting, robust stance recognition but fragile fine-grained matching due to semantic overlap and over-specificity, and polarity misclassification triggered by concession–rebuttal structures that mislead local cues. We also evaluated a range of models on the sub-tasks; the results make clear that current systems still have considerable room for improvement on pluralism-sensitive reasoning. Looking forward, we hope PERSPECTRA will serve not only as a benchmark but also as a foundation for new methods that explicitly engage with pluralism and perspectivism in language modeling, paving the way towards systems that move beyond single best answers and instead capture the diversity of reasoning, values, and perspectives that characterize human discourse.

## 7 LIMITATIONS

As limitations, first, the quality of opinion–comment retrieval is uneven as reflected in human evaluation; Reddit contains substantial noise and loosely related content, which can leak into expansions and reduce fidelity. Second, the benchmark currently centers on three sub-tasks (opinion counting, opinion matching, and polarity check), which capture important aspects but do not cover the broader space of pluralism. Third, manual validation was conducted on a limited subset of the corpus, leaving the need for larger and more diverse human evaluation.

## ACKNOWLEDGEMENTS

We gratefully acknowledge the support of the Lamarr Institute for Machine Learning and Artificial Intelligence. We also gratefully acknowledge the access to the Marvin cluster of the University of Bonn.

## REPRODUCIBILITY STATEMENT

The construction of PERSPECTRA is fully reproducible. All raw data are drawn from publicly accessible sources. The retrieval-and-expansion pipeline, including prompt templates, sampling parameters, and filtering rules, is released together with code in the supplementary material, ensuring that the enriched arguments can be regenerated. The three sub tasks-opinion counting, opinion matching, and polarity check—are derived deterministically from the dataset annotations, and the sampling procedure for evaluation instances is controlled with fixed random seeds. We provide scripts and configurations so that both dataset generation and task creation can be exactly reproduced.

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

## A  ETHICS STATEMENT

This work adheres to the ICLR Code of Ethics. In this study, no human subjects or animal experimentation was involved. All datasets used, including PERSPECTRA, were sourced in compliance with relevant usage guidelines, ensuring no violation of privacy. We have taken care to avoid any biases or discriminatory outcomes in our research process. No personally identifiable information was used, and no experiments were conducted that could raise privacy or security concerns. We are committed to maintaining transparency and integrity throughout the research process.

## B  LLM USAGE

Large Language Models (LLMs) were used to aid in the writing and polishing of the manuscript. Specifically, we used an LLM to assist in refining the language, improving readability, and ensuring clarity in various sections of the paper. The model helped with tasks such as sentence rephrasing, grammar checking, and enhancing the overall flow of the text. It is important to note that the LLM was not involved in the ideation, research methodology, or experimental design. All research concepts, ideas, and analyses were developed and conducted by the authors. The contributions of the LLM were solely focused on improving the linguistic quality of the paper, with no involvement in the scientific content or data analysis. The authors take full responsibility for the content of the manuscript, including any text go erated or polished by the LLM. We have ensured that the LLM-generated text adheres to ethical guidelines and does not contribute to plagiarism or scientific misconduct.

## C  APPENDIX: ANNOTATION CRITERIA FOR EXPANDED ARGUMENT EVALUATION

Each expanded version was evaluated along the following five dimensions, with scores ranging from 0 to 5.

**1. Fidelity to Original Opinion**  *Does the expanded version maintain the core argument of the original opinion?*

- **0** – Completely contradicts or misrepresents the original argument.
- **1** – Severely distorts the original argument with only minimal connection.
- **2** – Partially maintains the core argument but with significant alterations.
- **3** – Mostly maintains the core argument with some minor deviations.
- **4** – Maintains the core argument well with very minor deviations.
- **5** – Perfectly maintains the core argument with appropriate elaboration.

**2. Relevance of Added Content**  *Does the expanded version contain content that's unrelated to the original opinion?*

- **0** – Completely filled with unrelated content.
- **1** – Mostly unrelated content with minimal relevant parts.
- **2** – Contains significant unrelated content (roughly a half).
- **3** – Mostly relevant with some unrelated tangents.
- **4** – Very relevant with only minor unrelated elements.
- **5** – All added content is directly relevant to the original opinion.

**3. Use of Best Match Post**  *Does the model appropriately incorporate information from the best match Reddit post?*

- **0** – Completely ignored relevant best match OR fully incorporated irrelevant match.

- **1** – Made poor decisions about using the match (used very little when relevant OR used too much when irrelevant).
- **2** – Made questionable decisions about match usage.
- **3** – Mostly good decisions with some issues.
- **4** – Intelligently incorporated relevant match content OR appropriately avoided irrelevant content, with minor issues.
- **5** – Perfect discernment; fully utilized relevant information or completely avoided irrelevant match.

**4. Naturalness of Writing**  *Does the expanded version resemble human-written text?*

- **0** – Clearly machine-generated with obvious patterns, repetitions, or unnatural phrasing.
- **1** – Mostly machine-generated with many unnatural elements.
- **2** – Mixed quality with significant machine-like patterns.
- **3** – Generally reads like human writing with some artificial elements.
- **4** – Very natural, human-like writing with only occasional subtle artifacts.
- **5** – Indistinguishable from authentic human writing.

**5. Relevance of Best Match Post to the Topic**  *How well does the best match support the original opinion or relate to the topic?*

- **0** – Completely unrelated; no meaningful connection.
- **1** – Minimally related; shares only vague keywords or surface-level concepts.
- **2** – Weakly related; some abstract or tangential connection, but no direct alignment.
- **3** – Moderately related; touches on related issues or framing with partial support.
- **4** – Strongly related; clearly relevant or provides useful context.
- **5** – Directly supports; reinforces the original opinion or meaningfully engages with the topic.

## D  PROMPT USED FOR GENERATION

**Task Prompt Used for Expansion Generation**

> Your task is to expand the opinion given topics and posts or comments from Reddit that support this opinion. It doesn't need to be too long, just 3 to 4 sentences expansion. Ignore the post/comment if it's irrelevant, and try to mimic the wording style of the post/comment. Rewrite and expand the opinion in a natural, Reddit-style way.
> The topic is: {`topic`}.
> The opinion is: {`opinion`}.
> The post or comment is: {`comment`}.
>
> Please directly output the expanded opinion, don't include any other text.

## E  PROMPT USED FOR OPINION COUNTING

**Task Prompt Used for Opinion Counting**

> "Read the paragraph about {topic} and identify how many distinct opinions it contains. An opinion is a unique core stance; different wording with the same stance counts as one opinion. ONLY reply with an integer."

**Task Prompt Used for Opinion Matching**

"Read the following expanded paragraph about {topic} and decide which one of the listed original opinions it best matches. Reply ONLY with the index of the correct option (0-based)."

**Task Prompt Used for Polarity Check**

"Read the following concatenated opinions about {topic} and determine whether the majority stance is pro or con. Reply ONLY with 'pro' or 'con'."

## F  ERROR CASE FOR OPINION COUNTING

Table 5: An error case from the opinion counting task.

---

School is not enjoyable for many students because the pressure has ramped up significantly compared to previous generations. It's like people don't realize that what worked for them decades ago isn't necessarily going to work now. The workload is heavier, the competition is fiercer, and the expectations are sky-high. It's no wonder students are feeling overwhelmed and stressed out, yet some just brush it off as if it's the same experience they had ages ago.

Schools often prioritize conformity, pushing students to fit into a standardized mold that doesn't account for individual differences. This expectation can be particularly damaging in cultures with conservative attitudes toward mental health, where deviation from the norm is stigmatized. When students feel pressure to conform, they might suppress their unique identities and struggles, leading to increased stress and anxiety. The lack of support for mental health issues in such environments only exacerbates the problem, leaving students feeling isolated and misunderstood.

I hear you. It's not always the school itself that's the root cause of poor mental health; often, it's the environment and experiences surrounding it. Take bullying, for instanceŽ2014it's a huge factor that can make school a nightmare for kids, pushing them into online schooling just to escape the constant stress. It's not about the curriculum or the teachers, but more about how schools can sometimes fail to protect students from toxic interactions.

It's really unfortunate that sometimes things have to hit rock bottom before a child gets the help they need, but schools can be crucial in spotting those red flags early on. It's like, no matter where you are, mental health issues are everywhere, and it's frustrating because it feels like no one really knows how to handle it. But schools can step in and make a difference by identifying problems before they spiral out of control. They have the potential to be a place where kids can find support and resources, preventing those safeguarding issues from becoming a full-blown crisis.

Schools often play a crucial role in both identifying and addressing child safeguarding concerns, which can have a significant impact on a child's mental health. In many cases, schools are in a unique position to notice when something is off, like if a child has witnessed domestic violence or is showing signs of distress. Teachers and counselors can act as first responders, recognizing abnormal behavior and guiding the child towards getting the help they need. Without this support system, issues might go unnoticed, leading to long-term mental health problems that could have been mitigated with early intervention.

It's wild to think about how schools push everyone to fit into this one-size-fits-all mold, and if you don't, you're made to feel like an outcast. It's like you're constantly walking on eggshells, afraid to show who you really are because there's this looming fear of not being accepted. This pressure to conform can be incredibly damaging, especially when you're just trying to figure out your identity. It's no wonder so many students feel stressed and anxious when they're stuck in an environment that doesn't celebrate individuality.

Totally agree. Schools are like these factories where they expect every kid to fit into the same mold, and it really messes with their mental health. My daughter is already dealing with insecurities, and the pressure to conform just makes it worse. She's been bullied so much that we had to switch her to online school just to give her some peace. It's like the system doesn't accommodate individuality, and that can be really damaging for kids who are already vulnerable.

Absolutely, schools often push for conformity, and that can seriously mess with your mental health. I remember how my mom was super concerned that getting labeled would lead teachers to box me into this stereotype of being less smart or capable, even though she always saw my potential. It's like, if you don't fit into their neat little boxes, you're automatically seen as different, and that can make you feel out of place or even ashamed. Schools need to realize that not everyone fits the mold, and forcing us to do so can really take a toll on how we see ourselves and our abilities.

---

