# OpenReview forum: "PerSpectra: A Scalable and Configurable Pluralist Benchmark of Perspectives from Arguments"
_ICLR.cc/2026/Conference — ICLR 2026 Poster_

### Official Review · Reviewer_eQpS · 2025-10-26

**Soundness:** 2
**Presentation:** 3
**Contribution:** 2
**Rating:** 4
**Confidence:** 4

**Summary:**

This paper introduces a dataset (with associated benchmark tasks) for LLM alignment geared towards “pluralism,” i.e. the ability to engage with and distinguish between different perspectives on an issue. This is motivated by the idea that LLM outputs often reflect homogenized, culturally dominant opinions (thereby suppressing minority or contrarian views) and there is a relative lack of research on addressing this issue.
Their new dataset, called PERSPECTRA, draws from two online sources: Reddit, which has an abundance of unstructured debate text, and Kialo, which has very structured pro-con debate text. The dataset is constructed using NLP techniques: for each Kialo opinion, they automatically retrieve semantically similar Reddit comments and then, using controlled prompting with ChatGPT, generate expanded versions of the arguments therein.
The retrieval step is done in cosine similarity on the Qwen3-Embedding-8B model’s text embeddings. The expanded arguments are constructed by prompting GPT-4o “to preserve the core stance of the original Kialo opinion while adapting its style and elaboration using cues from the paired Reddit comment.” There are five expansions of each of the 762 opinions, yielding 3,810 expansions in total.

**Strengths:**

Overall, the paper seems strongest in terms of clarity.

Originality: This paper applies a relatively new method (LLM-aided retrieval-and-expansion) in a relatively understudied domain (pluralism alignment).
Quality: The analysis of failure modes is well done. In deploying a suite of LLMs on their benchmark, they are clear about the challenges faced by these models, and discuss the general trends about which models do better on which tasks.
Clarity: The paper and code seems very clearly written.
Significance: Their benchmark is well-motivated, tackling a legitimate problem in automated language generation––the (in)capacity for pluralism––which is of particular relevance as LLMs are used in educational and social contexts. Furthermore, though the dataset presented is somewhat small, the controlled retrieval-and-expansion pipeline suggests good scalability.

**Weaknesses:**

Overall, the paper seems weakest in terms of significance and originality.

Originality: The novelty of the methods involved in this paper––in particular the dataset construction––is not made very clear. Is there anything unique about the retrieval-and-expansion method used in this paper, aside from the choice of Reddit and Kialo as data sources?
Clarity: It is unclear to me how helpful Table 1 is. Is this supposed to show what the opinion counting task looks like? The caption should explain this. Also, the motivation in the introduction (e.g. the claim that LLMs output homogenized answers) should be fleshed out more. Seems like the content in Section 2.1 is relevant.
Quality: Scrolling through the LLM-expanded arguments contained in the dataset, I noticed that many of the arguments’ first sentences repeat the text of the topic word-for-word before going in more detail, while some start with simply “I agree.”
Significance: The benchmark is somewhat small (3810 arguments), compared to say PERSPECTRUM which has almost 20,000 combined claims, perspectives, and evidence sentences combined. This calls into question the claims of scalability. On a related note, I think using such a similar name to the existing PERSPECTRUM dataset should merit more discussion and comparison with that dataset.


Two more specific notes
Re: line 130-131: doesn’t PERSPECTRA, like PERSPECTRUM, rely on human annotation? Alternatively, if you believe the human annotation gives PERSPECTRA high enough marks, do you recommend creating a scaled up version of this dataset that is not subject to human annotation?
Re: Section 4/5: how do the suite of models perform on the other pluralism benchmarks? Are the trends discussed in the Failure Analysis section still representative?

**Questions:**

(1) Considering the claims of scalability made in the paper, I think it is natural to ask: why isn't PERSPECTRA bigger? Was the bottleneck the number of topics available on Kialo, or the number of human evaluators?
(2) A key part of the dataset creation is asking ChatGPT to expand on the opinion given in a Reddit comment. Is there a risk that asking an LLM to expand on a comment is not so valid, depending on the nature of the comment? Is this an issue you ran into at all?
(3) In the process of matching Kialo opinions with Reddit comments, was there a worry that some of the Reddit comments were bot-generated? It seems like there was some filtering for this (re: line 197), but it seems like comments written by a chatbot which don’t have bot in the username could still reasonably make it past your filtering?
(4) Who performed the manual validation? The authors? External party? How many people?
(5) I’m not sure how much Figure 3 contributes to the main text of the paper. Is it communicating anything unique about the dataset? Is there a particular statistic here that merits attention? I suppose this may just be a standard thing to do for papers which present a new dataset.

---

> ### Author Response · Authors · 2025-11-21
>
> We thank you for the insightful remarks and for commending our new method in an understudied domain.
>
> Our responses are as follows:
>
> **W1: Originality**
>
> Retrieval-and-expansion itself is not our novelty. PERSPECTRA’s contribution is introducing **structured argumentation** into pluralism research: Kialo’s hierarchical topic–opinion graph provides the grounded stance-level structure needed for tasks such as opinion counting and fine-grained matching, which existing datasets cannot support.
>
> **W2: Clarity**
>
> Table 1 is meant to illustrate both the input format for opinion counting and the reasoning challenge it poses: models must **navigate** multiple, interleaved opinions and **identify** which sentences express the same underlying viewpoint, despite lexical and stylistic variation. We have updated the caption in Tabel 1 in the revised version.
>
> **W3: Quality**
>
> Thank you for the observation. In the current dataset, expansions beginning with “I agree” occur only **5 times out of 3,810** total examples, indicating that such patterns are rare and not systematically produced.
>
> **W4: Significance**
>
> Scalability here refers to **coverage and ease of extension**: once a structured topic–opinion set exists, the pipeline generates expansions **without human annotation**. The human evaluation in our paper only audits a small subset; **the dataset itself does not rely on annotation**.
>
> The observed failure modes arise because our tasks require **multi- perspective semantic disentanglement**, which other pluralism benchmarks (survey-based or safety-norm datasets) do not support.
>
> **Q1:**
>
> as explained in Significance
>
> **Q2:**
>
> Yes, there is this problem, but the model used for opinion expanding(chatGPT4o) is just one of the test models.
>
> **Q3:**
>
> Indeed, our current filter only removes comments from users that have “bot” in the username,  but this limitation is shared widely across social-media–based datasets, where reliable bot detection remains an open challenge.
>
> **Q4:**
>
> Two graduate students studying NLP performed human annotation on a subset of the dataset.
>
> **Q5:**
> Yes, you are right, we think a dataset paper should report such statistics.

---

### Official Review · Reviewer_nZ2F · 2025-10-28

**Soundness:** 2
**Presentation:** 3
**Contribution:** 3
**Rating:** 6
**Confidence:** 4

**Summary:**

The authors propose PERSPECTRA, a benchmark for evaluating the "pluralism" of LLMs, which they define as their ability to handle multiple distinct viewpoints without collapsing them. The core contribution is a synthetic data pipeline: they take structured arguments from Kialo, retrieve semantically "similar" comments from Reddit, and use GPT-4o to expand the arguments into naturalistic text that mimics the Reddit style. From this dataset of 3,810 arguments across 100 topics, they define three tasks: counting unique opinions, matching a generated argument back to its source, and checking the aggregate pro/con polarity. They benchmark several models, highlighting common and interesting failure modes like overestimating opinion counts and being tricked by concessive phrasing.

**Strengths:**

The core idea of combining the structural clarity of Kialo with the stylistic diversity of Reddit is a neat trick. It's a pragmatic approach to generating nuanced argumentative data, which is a known bottleneck. I wasn't even aware of Kialo before this paper.

The real meat of the paper is in Section 5. The identification of challenges like "opinion overestimation" and the "concession trap" is good.

The idea of creating a benchmark for pluralism is good, and the generation method is a clever (if flawed) approach to scale beyond human annotation. The benchmark and the failure analysis are a welcome addition, even if the dataset is not massive.

The paper is clearly written. The pipeline is explained well, and the failure analysis section is particularly insightful.

**Weaknesses:**

The paper champions a "scalable" pipeline but delivers a dataset with only 100 topics and 3810 rows. This feels more like a proof-of-concept than a large-scale resource that lives up to the "scalable" moniker. I'd call OpenDebateEvidence and it's nearly 4 million rows "scalable", but not this.

The related work section is incomplete. It fails to cite foundational work in this niche, particularly datasets like DebateSum and OpenDebateEvidence (Roush et al.). These works tackle the problem of structuring claims from formal debate rounds and are highly relevant to this papers related work sections. I would also expect similar citations for IBM Project Debater and the things they cite.

The entire benchmark is built on outputs from a single proprietary model (GPT-4o). Are we testing a model's understanding of pluralism, or its ability to parse the stylistic tics of another LLM? This creates a "synthetic-on-synthetic" evaluation loop that risks optimizing for model-to-model mimicry rather than genuine reasoning. It's the hall of mirrors problem, and the paper doesn't sufficiently address it.

The authors' own human evaluation reveals the retriever is mediocre. If the retrieved Reddit comment is irrelevant, then the claim of "integrating the linguistic diversity of real Reddit discussions" falls flat. The generation becomes simple paraphrasing-with-a-prompt rather than a true synthesis.

**Questions:**

Given the low relevance score for the retrieved Reddit comments (3.31/5), how can you be sure the final dataset genuinely captures Reddit's linguistic diversity, rather than GPT-4o's simulation of it? Could you provide an analysis of how much the generator actually uses the retrieved comment, especially in cases where its relevance is low?

Have you considered the potential for evaluation contamination, where models are benchmarked on data generated by a close relative (or even the same underlying architecture)? Does this risk rewarding models that simply mimic the generator's style over models that perform genuine semantic reasoning?

---

> ### Author Response · Authors · 2025-11-21
>
> We appreciate your careful reading and valuable suggestions.
> Below we clarify the points raised.
>
> **W1: claim of “scalable”**
>
> Our claim of scalability refers to the pipeline, fully automatic and free from human annotation, not the current provided dataset size. We released 100 topics and 7.6 opinions on avg for each topic because we selected a representative subset for this version, not because the pipeline has an upper limit. As more topics are included from Kialo, the same procedure can generate additional expansions and tasks without any manual effort.
>
> **W2: Missing related work**
>
> Thank you for the related work suggestions! We have incorporated the mentioned works in section 2.3 with highlights (lines 132- 138 and 142-144) in the revised version.
>
> **W3: Synthetic-on-synthetic concerns**
>
> Although GPT-4o produces the expansions, the core semantic structure comes from Kialo opinions, and the stylistic cues come from human-written Reddit comments. The three tasks rely on semantic grounding (unique opinion IDs, stance labels) rather than surface style. Empirically, several open-source models outperform GPT-4o on Table 4 in Section 5, indicating that the benchmark does not reward generator-specific artifacts.
>
> **W4: Low retriever relevance score and Q1:**
>
> The 3.31/5 score reflects **topical relevance**, not the presence of **Reddit linguistic markers**. In practice, even partially relevant comments contain distinctive Reddit phrasing, which the generator incorporates. Our human evaluation shows high scores for “use of best match,” meaning the model selectively uses useful cues and ignores noise.
>
> As far as we understand, it is difficult to accurately measure the linguistic diversity and we are not aiming to propose new metrics for this, as it is out of the scope of this paper. We are more than happy to include existing methods/metrics to measure the diversity. Please let us know if there are any specific metrics that you would expect to see!
>
> **Q2:**
>
> Our results suggest no such bias: open-source models (e.g., QwQ-32B, Qwen models) outperform GPT-4o on two of the three tasks (Table 4 in Section 5). Since ground truths depend on symbolic stance structures rather than generative style, the benchmark evaluates pluralistic reasoning rather than mimicry.

---

> > ### Comment · Reviewer_nZ2F · 2025-11-26
> > **I have raised my score**
> >
> > Thank you for the modifications and further clarification. I have raised my score.

---

### Official Review · Reviewer_uHvT · 2025-10-31

**Soundness:** 3
**Presentation:** 4
**Contribution:** 3
**Rating:** 8
**Confidence:** 4

**Summary:**

The authors propose PerSpectra, a large-scale, curated dataset combining Kialo's debate structure, which contains pro and con opinions on 100 different topics, and comments from Reddit. This dataset is extended with statements generated by large language models (LLMs). They also introduce three tasks that can be solved using this dataset: opinion counting, opinion matching, and polarity checking.

**Strengths:**

The paper is well-written and easy to follow. The data curation process and the proposed tasks and metrics are clearly described and useful for comparing the capability of large language models to recognize plurality of opinions.

**Weaknesses:**

The authors claim that this dataset differs from others that focus on pluralistic opinions on various topics because the process did not require extensive human annotation. However, annotating a sub-sample revealed the possibility of selecting statements that do not fit the topic, which compromises the data's quality. Human annotation is therefore indispensable. Nevertheless, this dataset is a good starting point for further research and offers an interesting approach to obtaining debate data.

**Questions:**

No questions left.
Small comments:
- 042: The abbreviation for LLMs is introduced in the Related work section, but should already be introduced in the Introduction.
- 059, 062-063: Some of the citations in the introduction that are written out in the text should be put in parentheses using the command "\citep."
- 119: maybe you meant LLMs instead of LMs?
- 244: Small typo in "Writting" (instead of Writing)

---

> ### Author Response · Authors · 2025-11-21
>
> We thank you for the positive assessment and valuable feedback.
>
> **W1:**
>
> We would like to clarify that our human evaluation was conducted on a randomly sampled subset of the dataset, without selecting for quality or topic relevance.
>
> All mentioned minor errors are addressed in the revised version of the paper.

---

### Official Review · Reviewer_A1ew · 2025-10-31

**Soundness:** 2
**Presentation:** 2
**Contribution:** 2
**Rating:** 4
**Confidence:** 4

**Summary:**

- Defines pluralism as the ability to represent multiple valid perspectives.
 - Argues that debate-oriented data (Reddit, Kialo) naturally capture pluralism but each has limitations: Reddit is linguistically diverse yet unstructured, while Kialo is structured but too concise.
 - Introduces Perspectra, a benchmark that merges Reddit’s linguistic diversity with Kialo’s argument-graph structure.
 - Establishes three evaluation tasks:
	- Opinion counting — detecting distinct viewpoints.
	- Opinion matching — aligning discourse to original stances.
	- Polarity check — inferring the overall stance in mixed text.

**Strengths:**

LLMs are increasingly used in sensitive contexts and naturally plurality of opinions is inherent to the problem. We need a lot of work to define new goal posts in this direction, like this work.

**Weaknesses:**

The "Opinion Counting" (estimating how many distinct opinions are present within a paragraph) is a somewhat ambiguous. Like you the definitions of what constitutes as a distinct opinions can be subjective and can vary from one person to another. The same also holds for the other tasks (matching and polarity).

In other words, why should I (or any reader) trust that:
 - you have high-quality annotations?
 - and that the task is well-defined?

I do see that you conduct human annotations to ensure data quality. Have you tried computing the human performance for each of your tasks? (counting, matching, polarity). Having these numbers can help inform the reader about data quality vs ambiguity.

**Questions:**

I'd like to hear your perspective on other benchmarks that have been release on pluralistic alignment.

 - [1] CoSApien https://huggingface.co/datasets/microsoft/CoSApien https://arxiv.org/abs/2410.08968
 - [2] MOREBENCH https://huggingface.co/datasets/morebench/morebench https://www.arxiv.org/abs/2510.16380
 - [3] https://huggingface.co/datasets/facebook/community-alignment-dataset  https://arxiv.org/abs/2507.09650

---

> ### Author Response · Authors · 2025-11-21
>
> Thank you for your time and valuable suggestions.
> Below are our responses.
>
> **W1: All three subtasks are subjective**
>
>
> We agree that pluralistic-related tasks naturally involve subjectivity. However, PERSPECTRA minimizes annotator subjectivity by **not defining opinions ourselves**: all opinion units come from **Kialo’s moderated debate graphs**, where topic–opinion boundaries are already curated and stable. Annotation therefore only **maps text to fixed opinion nodes**, rather than deciding what counts as an opinion.
>
> The presence of **multiple expansions per opinion** further provides internal consistency, within-opinion statements cluster tightly, while cross-opinion statements diverge. To handle residual ambiguity, our evaluation uses **graded, distance-based metrics** (MAE, NIE) instead of just brittle exact-match accuracy, ensuring proportional penalties rather than all-or-nothing judgments.
>
> Together, these design choices make the tasks well-defined and the annotations reliable despite the inherent subjectivity of pluralism. We have updated the text accordingly (lines 373 to 377) in the revised version.
>
> **W2: Human performance on subtasks**
>
>
> This is an excellent point. For the rebuttal, we added human performance on the Opinion Matching subtask (In Table 4), which we think can be completed within the current timeframe. We will apply the same evaluation procedure to the remaining subtasks and include full human-performance results in the camera-ready version.
>
> **Q1: Our perspective on other benchmarks**
>
>
> CoSApien (Zhang et al., 2025), Community Alignment (Zhang et al., 2025), and MOREBENCH (Chiu et al., 2025) each operationalize a different facet of pluralistic alignment.
>
> CoSApien targets **normative-safety pluralism**, using expert-defined configurations but covering a narrow set of safety-critical contexts.
>
> Community Alignment captures **end-state preference pluralism**, and MOREBENCH captures **process-level preference pluralism**; both are primarily used as **training data** for preference-based alignment rather than as structured multi-viewpoint evaluation benchmarks.
>
> Importantly, none of these resources provide settings in which multiple, competing **opinions on the same topic** are explicitly defined or jointly represented. They do not attempt to identify or separate such distinct opinions.
>
> In contrast, PERSPECTRA addresses **perspective-level pluralism**. Opinion identities are inherited directly from **Kialo’s curated debate graph**, where competing viewpoints are already explicitly differentiated. We then generate multiple user-like realizations for each  opinion via GPT, creating rich within-opinion variation while preserving clear opinion-level boundaries.
>
> While CA or MOREBENCH could in principle be reformulated into stance-style tasks, they lack this explicit argumentative hierarchy, making them unsuitable for evaluating multi-opinion reasoning.
> PERSPECTRA therefore complements these resources by providing a distinct evaluation setting for perspective-level pluralism.
>
> We have added these datasets in the related work in Section 2.2 with highlights (line 115-122) in the revised version.

---

> > ### Comment · Reviewer_A1ew · 2025-11-24
> >
> > > We agree that pluralistic-related tasks naturally involve subjectivity.
> >
> > Glad we agree.
> >
> >
> > > all opinion units come from Kialo’s moderated debate graphs, where topic–opinion boundaries are already curated and stable.
> >
> > How do you know that we can trust the Kialo annotations?
> >
> >
> > How do you know an opinion only convey's the content of a particular opinion and not the adjacent/related opinion?
> > Basically, the question is less about the latest argument structure and more about how you'd slice and map a given paragraph (expanded comment) into distinct arguments (even if you trust this anotation). This is inherently subjective.
> >
> > > Q1
> >
> > These are reasonable. Thanks!
> >
> >
> >
> > * The definition of the "counting" task is also a bit silly, IMO (who cares about counting the arguments.) If the goal is to measure "which arguments appear in this paragraph", why not measure that directly? This is the "matching" task which I think should be the target.
> >
> >  * I see the "human" at the last row of table 4. If "human" is not evaluated for certain columns, you should put NA or -- rather than 0.0.
> >
> >  * Kialo data has tree structure and can have more than 2 opinions. Do you make use of it? The exact in 3.1 seems to show one opinion for pro vs one opinion for con only.

---

> ### Author Response · Authors · 2025-11-26
>
> Thank you for your feedback!  Below are our following responses:
>
> **Clarification on Opinion Annotation and Expansion**
>
> Kialo topics are curated by designated **owners and admins** who decide which claims are accepted, where they are placed, and whether they support or oppose their parent claim.
> User submissions are treated as suggestions; moderators approve, reposition, or reject them to ensure that each expresses a single, focused argument.
> Redundant claims are **flagged by automatic duplication warnings** and are resolved through **merging or linking by moderators**, resulting in a consistently structured debate tree with clearly separated opinion paths.
> While Kialo does not label opinions in the formal academic sense, its combination of granular claims, stance tagging, and human moderation provides a stable basis for opinion-level annotation.
> We acknowledge that expansions may broaden the semantic scope of the original claim. However, Kialo’s opinions are typically short (often a phrase or single sentence) and lack sufficient expressive variation. To generate more naturalistic arguments, we prompt an LLM to expand each opinion using a topically matched Reddit comment (retrieved via embedding similarity) as auxiliary context. This yields diverse, human-like expressions while maintaining stance fidelity **without human selection or filtering**.
>
> **Clarification on the Purpose of the Counting Task**
>
> This task also complements opinion matching by targeting a **different dimension of opinion understanding**. While opinion matching evaluates whether a model can link an expanded argument back to its source opinion, counting assesses whether the model can **discern conceptual boundaries between expansions**—i.e., distinguish which expansions correspond to the same underlying opinion and which belong to separate ones. In this sense, matching addresses **opinion-to-expansion grounding**, whereas counting probes **expansion-level differentiation** across overlapping argumentative content.
>
> **Clarification on Opinion Coverage and Tree Structure**
>
> Yes, we do make use of the full first-level tree structure in Kialo. For each topic, we include **all available top-level opinions**, not just one Pro and one Con. The example in Section 3.1 was simplified to illustrate our data format. As shown in Table 3b, our dataset includes an average of 3.71 opinions for pros and 3.89 for cons.
>
> **NA on human performance**
>
> Thank you for pointing that out. We have uploaded a new version, and will put the full human performance in days.

---

### Author Response · Authors · 2025-12-02
**Summary of Rebuttal**

We sincerely thank the reviewers for their thoughtful feedback. Below we provide a concise summary of the major points raised and our corresponding clarifications and revisions. Each entry highlights the core concern, the reviewers who raised it, and our consolidated response. For full details and line-level updates, please refer to the complete rebuttal section.

| Issue     | Description |Raised by|Explanation or Modification|
| - | - |-|-|
| Claim of Scalability | Reviewers questioned whether a dataset of 100 topics / 3810 rows is “scalable” |eQpS, nZ2F|Scalability refers to the **pipeline itself**. The retrieval–expansion workflow is **fully automatic** and can be applied to any number of Kialo topics **without human annotation**, providing both broad **coverage** and **ease of extension**. The current dataset is a representative slice, not a limit.|
| Missing Related Works   | Reviewers noted missing citations to debate-related corpora and pluralism-related benchmarks |A1ew, nZ2F|The mentioned works have been added to the related work section, and we clarified how PERSPECTRA differs by providing **explicit multi-opinion structure** that supports **fine-grained pluralistic reasoning tasks**.|
| Data Quality and Human Eval  |  Concerns about small-scale human eval, potential cherry-picking, artifact patterns and low retrieve relevance score. |A1ew, nZ2F, eQpS, uHvT|Human-eval samples were **uniformly random**, and artifacts are **extremely rare (5/3810)**. **Full human performance** across sub-tasks is included in the revised version. The retrieval relevance score reflects **topical alignment** rather than Reddit style; even partially related comments still provide **Reddit phrasing**, and high “use of best match” scores show that the generator **selectively incorporates** useful cues while ignoring noise.|
| Subjectivity and definitions of Sub-Tasks | Distinct-opinion boundaries may be subjective; counting task seems less meaningful than matching. |A1ew|Pluralism-related tasks inevitably involve some subjectivity, but our setup introduces no annotator discretion by **inheriting all opinion units from Kialo’s moderated debate graphs**, so annotation only maps texts to fixed opinion nodes rather than defining opinions. **Multiple expansions** per opinion and the use of graded, **distance-based metrics** (MAE, NIE) instead of brittle exact matches further improve consistency and robustness to residual ambiguity. Opinion matching and opinion counting capture different layers of opinion understanding: matching evaluates **opinion-to-expansion grounding**, whereas counting probes **expansion-level differentiation** across overlapping content.|
| Reliability of Kialo Opinion Boundaries | Reviewers questioned whether Kialo’s opinion nodes are reliable anchors |A1ew|Kialo’s opinion structure is actively curated by topic owners and moderators: user-submitted claims are **reviewed, approved, merged, or repositioned**, with duplication warnings and stance tagging ensuring that each node expresses a single, focused argument. This yields a consistently structured debate tree with clear opinion boundaries. Our expansions add naturalistic variation using matched Reddit context while preserving stance fidelity, providing linguistic diversity without altering the underlying opinion boundaries.|
| synthetic-on-synthetic | Expansion uses GPT-4o; evaluation may reward models that mimic its style. |nZ2F|Although GPT-4o produces the expansions, the core semantic structure comes from Kialo opinions, and the stylistic cues come from human-written Reddit comments. The three tasks rely on semantic grounding (unique opinion IDs, stance labels) rather than surface style. Empirically, several open-source models outperform GPT-4o on Table 4 in Section 5, indicating that the benchmark does not reward generator-specific artifacts.|

---

### Meta-Review · Area_Chair_7UF5 · 2026-01-19

**Summary:**

This paper proposes PERSPECTRA, a benchmark for evaluating the "pluralism (ability to handle multiple distinct viewpoints)" of LLMs. The key contribution is a synthetic data pipeline: they take structured arguments from Kialo, retrieve semantically "similar" Reddit comments, and use GPT-4o to expand the arguments into naturalistic text that mimics the Reddit style but maintains the Kialo opinion's stance. The dataset is composed of 3,810 arguments across 100 topics that evaluate models on three tasks: opinion counting, opinion matching, and checking the aggregate pro/con polarity. The evaluation of several models highlights common failure modes, such as overestimating opinion counts and being tricked by concessive phrasing.

**Reviewer Concerns:**

- The claim of "scalable" is not supported by the relatively small dataset size when compared with existing plurism benchmarks.
- The annotations and tasks can be subjective and biased, as Kialo annotations and Reddit comments are also written by humans (or even bots).
- The paper does not discuss various related works.
- The risk of "synthetic-on-synthetic" and evaluation contamination: the arguments are expanded and generated by GPT-4o and are then used to evaluate LLMs.
- The quality of some LLM-expanded arguments is low.
- Lack of side-by-side comparison with the evaluation results on other plurism benchmarks.
- The originality and novelty of the data pipeline are limited.
- The manual validation is done by two graduate students on a small subset, which might not be sufficiently reliable.

**Reviewer Scores:**

- The initial reviewer rating is 4, 4, 6, 8, with uniform confidence of 4.
- Two reviewers with ratings of 4 participated in the discussion after rebuttal, one of them raised the rating from 6 to 8.
- The authors did a good job responding to the first three reviewers but did not provide a concrete response to the last one (with a rating 4).
- I carefully checked the paper and discussion. I think some concerns have been addressed. However, some major ones, e.g., subjectivity of annotations, scalability, human validation, quality and bias of synthetic arguments, are not fully addressed (and might be challenging to resolve satisfactorily).
- That being said, the benchmark is novel in providing a perspective-level pluralism evaluation suite. Considering the review ratings, I recommend acceptance.

---

### Decision · Program_Chairs · 2026-01-26

Accept (Poster)